# Meta-regression to explain shrinkage and heterogeneity in large-scale replication projects

**Rachel Heyard**[ID]*, **Leonhard Held**

Center for Reproducible Science, Epidemiology, Biostatistic and Prevention Institute, University of Zurich, Zurich, Switzerland

* rachel.heyard@uzh.ch

## Abstract

Recent large-scale replication projects (RPs) have estimated concerningly low reproducibility rates. Further, they reported substantial degrees of shrinkage of effect size, where the replication effect size was found to be, on average, much smaller than the original effect size. Within these RPs, the included original-replication study-pairs can vary with respect to aspects of study design, outcome measures, and descriptive features of both original and replication study population and study team. This often results in between-study-pair heterogeneity, i.e., variation in effect size differences across study-pairs that goes beyond expected statistical variation. When broader claims about the reproducibility of an entire field are based on such heterogeneous data, it becomes imperative to conduct a rigorous analysis of the amount and sources of shrinkage and heterogeneity within and between included study-pairs. Methodology from the meta-analysis literature provides an approach for quantifying the heterogeneity present in RPs with an additive or multiplicative parameter. Meta-regression methodology further allows for an investigation into the sources of shrinkage and heterogeneity. We propose the use of location-scale meta-regressions as a means to directly relate the identified characteristics with shrinkage (represented by the location) and heterogeneity (represented by the scale). This provides valuable insights into drivers and factors associated with high or low reproducibility rates and therefore contextualises results of RPs. The proposed methodology is illustrated using publicly available data from the Replication Project Psychology and the Replication Project Experimental Economics. All analysis scripts and data are available online.

## Introduction

In the last decade, numerous large-scale replication projects (RPs) were conducted to assess the reproducibility in, among others, the research fields of Psychology [1], Experimental Economics [2], Social Sciences [3], Experimental Philosophy [4] and Cancer Biology [5]. These projects selected a set of highly cited or influential papers in well established journals and

**Data availability statement:** The code and data to reproduce our analyses are openly available from our institutional gitlab via https://gitlab.uzh.ch/rachelheyard/heterogeneity_in_replication_projects. A citable

snapshot of the repository was uploaded to Zenodo and is available via the following DOI: https://doi.org/10.5281/zenodo.10590041. The data used in our case study was directly downloaded from the OSF project "Predicting the Replicability of Social Science Lab Experiments" (https://osf.io/4fn73/), available via https://osf.io/vfhcw.

**Funding:** The author(s) received no specific funding for this work.

**Competing interests:** The authors have declared that no competing interests exist.

attempted direct replications of the main results from the original studies. A setup where all selected original studies are replicated once is refered to as "Many Phenomena, One Study" [6]. A direct replication is defined as a study "following the methods presented in the original research as close as possible to retrieve new data and achieve consistent results using the same statistical analysis" [7]. The Replication Project Psychology [RPP,1] concluded that while 97% of the 100 original studies reported a significant result (with $p$-value <0.05), only 36% of the replication studies found a significant result with effect estimate in the same direction. 39% of the effects were subjectively rated by the replication teams to have replicated the original result. The Replication Project Experimental Economics [RPEE,2] was a much smaller effort compared to the RPP, with only 18 original-replication study-pairs. Here, all included original findings were statistically significant but only 61% of the replications found a significant effect in the same direction as the original study. The replication effect sizes were found to be significantly smaller in absolute value than the original effect sizes. This phenomenon is commonly known as *shrinkage* and has been observed in all large-scale replication projects mentioned above [8,9]. Shrinkage in large-scale RPs has been attributed to a tendency of the original effect estimates being inflated due to publication bias or questionable research practices [10]. Further, between-study-pair *heterogeneity*, i.e. variation in effect size differences across study-pairs that goes beyond expected statistical variation, is often observed in RPs but generally ignored when the results are summarized. This led some to criticize the methodology employed by RPs as they tend to draw negative conclusions with respect to the reproducibility of the investigated research fields and their practices [6,11]. Shrinkage and heterogeneity are directly linked to reproducibility rates and, while a certain degree of heterogeneity and shrinkage might not be avoidable, identifying covariates associated with high or low levels can put low reproducibility rates into context. In the following, we begin by discussing how heterogeneity can be assessed and how methods from meta-analysis have been applied in the analysis of RPs. We then introduce location-scale meta-regression as a tool to relate specific covariates to both shrinkage and heterogeneity in oder to draw more nuanced interpretations of the results from RPs.

**Assessment of heterogeneity in the meta-analysis literature.** The concept of heterogeneity between studies is well understood in the meta-analysis literature [12–14]. Most published meta-analyses test for between study heterogeneity and, if needed, account for it using, for example, a random-effects meta-analysis [15]. Heterogeneity can be quantified either using an additive or a multiplicative model. The additive model or random-effects meta-analysis, as implemented in the `metafor` R-package [16], is commonly employed in the meta-analysis literature and accounts for heterogeneity by adding a constant $\tau^2$ to each study's variance. The multiplicative version, on the other hand, relies on a weighted linear regression model with weights equal to the inverse of the studies' variances. The effect size returned by this multiplicative model is the same as the one from a fixed effect meta-analysis with variances multiplied by a constant $\varphi$, the multiplicative heterogeneity parameter [17]. The additive and multiplicative model mainly differ in their assumptions on the underlying effect sizes. The multiplicative model shares the same assumption of a single overall effect size with the traditional fixed effect meta-analysis. The additive model allows, like the random-effects meta-analysis, for some variability in the true effect sizes [18].

Once heterogeneity using either model version has been established and quantified, meta-regression can be used to investigate whether covariates account for part of the heterogeneity in effect sizes. As explained in Chapter 7 of Schmid et al. (2020) [19], a meta-regression is conceptually the same as a traditional (weighted) regression, but instead of individual subjects, the units of analysis are studies. Such meta-regression models allow researchers to

directly add continuous and/or categorical covariates into a model to investigate the covariates' association with the effect size [12]. The remaining variability that is not accounted for by the included covariates is the *residual heterogeneity*, $\tilde{\tau}^2$ and $\tilde{\varphi}$ respectively. Standard meta-regressions, both additive and multiplicative versions, assume that the amount of residual heterogeneity remains constant across studies. To overcome this assumption and allow heterogeneity to be dependent on covariates and consequently be specific for the studies themselves, Viechtbauer and López-López (2022) [20] suggest to use location-scale meta-regression which further allows researchers to investigate which covariates are associated with the amount of heterogeneity. Model selection to reduce the risk of overfitting can be readily applied for location-scale meta-regression [21].

**On the use of meta-analysis methods for the analysis of large-scale RPs.** Methodology from the meta-analysis literature has been used to quantify the reproducibility of findings or assess successful replication. For this, results from the original and replication studies are pooled together using a fixed effect meta-analysis. However, significance of the combined effect estimate as a "replication success" metric has limitations, as it will almost certainly flag success if the original study result is very convincing (small *p*-value) even if the replication *p*-value is large [22]. Since there is no universally agreed-up on criterion for replication success, most large-scale replication projects used a whole set of metrics [23]. The RPP, for example, used significance and *p*-values, effect sizes, subjective assessment of the replication teams, and meta-analyses of effect sizes as metrics for replication success and to compute overall reproducibility rates. Even though replication studies usually follow a strict protocol that adheres as close as possible to the original study, it is unavoidable that slightly varying conditions in the two studies can lead to variability in the underlying true effects. Sources of such within-study-pair heterogeneity are manifold (see for example Table 1 in Bryan et al. (2021) [24] for behavioral intervention research) and include differences in the study population or in the definition of the outcome or intervention. For example, many replication studies sample participants from slightly different populations [11]. Of the previously mentioned large-scale RPs, only the RPCB discussed within-study-pair heterogeneity and accounted for it in sensitivity analyses by re-estimating the agreement in significance across study-pairs under the assumption that there was within-study-pair heterogeneity [5]. To estimate the amount of heterogeneity they had to use preregistered many-lab experiments in Psychology [25], because within-study-pair heterogeneity cannot be estimated in the standard "Many Phenomena, One Study" setup. Many-lab experiments follow a "One Phenomenon, Many Studies" setup, attempting to directly replicate one original finding many times in different labs or groups [6]. A key goal of these many-lab experiments is to examine the heterogeneity of effect size within replication studies [26,27].

Turning back to large-scale RPs, Pawel and Held (2020) [10] used a model which can take into account shrinkage and within-original-replication-study-pair heterogeneity to predict the replication effect estimate. They concluded that some degree of heterogeneity between original and replication effects should be expected. In a related setting, Röver and Friede (2024) [28] used methods from meta-analysis to investigate heterogeneity in pairs of so called "study twins", two similar confirmatory clinical phase III trials that are based on a common protocol. They concluded that a single study-pair "provides only very little evidence on the heterogeneity" within the pair.

Summarising the results across all original-replication study-pairs included in the same RP into an overall reproducibility assessment of a research field, as usually done in RPs, is challenged by the presence of between-study-pair heterogeneity. Meta-regression methodology could help to contextualize the results of heterogeneous projects, but has only rarely been

used to examine the results of reproducibility efforts. The RPP team used meta-regressions to investigate evidence for publication bias in their data, by first testing whether the original, respectively replication, standard errors are associated with the original, respectively replication, effect sizes, and second, whether the original standard error is associated with the difference in effect sizes [1,Supplement Sect 4.g and 4.h]. Bench et al. (2017) [29] reanalysed the results from the RPP and used meta-regression to investigate whether the expertise of the replication team influenced the results of replication studies (i.e., replication effect size). In another reanalysis of the RPP data by van Bavel et al. (2016) [30], regression models were used to examine the association between reproducibility (measured by the binary rating that the original results were successfully replicated) and contextual sensitivity. Altmejd et al. (2019) [31] used linear regression to predict the relative effect size (replication effect size divided by original effect size) using parameters related to the design and properties of the author team of the original and replication studies, with data from RPP, RPEE and two many-lab experiments.

**Explaining shrinkage and heterogeneity using location-scale meta-regression.** Location-scale meta-regressions should be well suited to help comprehend what the primary drivers of shrinkage (i.e., the location) and between-study-pair heterogeneity (i.e., the scale) are. More specifically, when planning future replication projects, it is valuable to know what levels of shrinkage and amount of heterogeneity in effect size differences are to be expected as this directly influences the reproducibility rate estimated in the project, which is often based on effect size differences. When analysing the results from large-scale RPs, researchers might want to perform reproducibility subgroup analyses by estimating the reproducibility separately for those original-replication study-pairs with specific characteristics linked to the location and/or the scale.

The additive version of the (location-scale) meta-regression is more broadly used, but the multiplicative version with its assumption of a single overall effect size, might be particularly well suited in the replication setting, where replication studies are based on study protocols that are very similar to those of the original studies [17,Sect 4]. Recently, multiplicative meta-regressions have been employed to explore how design differences are associated with the variation in results between study-pairs of randomised trials and their replications performed with real world data [23]. The difference in effect sizes, which should be close to zero if there were no shrinkage, is used as outcome variable and the multiplicative between-study-pair heterogeneity can be readily extracted from this model.

In the present paper, we use and extend methodology from the meta-analysis literature to study shrinkage of effect size and heterogeneity in effect size differences between original-replication study-pairs in the research fields Psychology and Experimental Economics. Location-scale meta-regression will help identify potential sources of shrinkage and heterogeneity, and allow for a more nuanced conclusion on the reproducibility of the research published in those fields. The proposed methodology is presented in the following section and illustrated in a case study. We close with a discussion of the results and the limitations.

## Methods

### Statistical assessment of large-scale replication projects

A replication project is composed of $n$ independent original-replication study-pairs. The difference in effect size between an original study $i$ and its replication – the outcome of interest – is defined as $\Delta_i = \theta_{oi} - \theta_{ri}$, where $\theta_{oi}$ and $\theta_{ri}$ are the underlying effects, estimated by $\hat{\theta}_{oi}$ and

$\hat{\theta}_{ri}$, in either the original study or the replication study. Note that $i \in \{1, \ldots, n\}$ and the difference $\Delta_i$ is estimated by $\hat{\Delta}_i = \hat{\theta}_{oi} - \hat{\theta}_{ri}$. The effect size can be a mean difference, a log odds ratio, a correlation or similar and might have to be transformed to follow approximately a normal distribution [32]. The effect type of all $n$ study-pairs is assumed to be the same, by default or after transformation, and the original effects are all oriented to be positive. The standard errors are denoted by $\sigma_{oi}$ and $\sigma_{ri}$ respectively. Original and replication study are assumed to be two independent studies, each collecting its own data, and the standard error of the effect size difference will be $\sigma_{\hat{\Delta}_i} = \sqrt{\sigma_{oi}^2 + \sigma_{ri}^2}$. An important quantity in the assessment of replication projects is the standardized difference [33], which measures compatibility of effect sizes:

$$\hat{\delta}_i = \frac{\hat{\theta}_{oi} - \hat{\theta}_{ri}}{\sqrt{\sigma_{oi}^2 + \sigma_{ri}^2}} = \frac{\hat{\Delta}_i}{\sigma_{\hat{\Delta}_i}}. \tag{1}$$

Under the null hypothesis $H_0$ of homogeneity between studies, $\hat{\delta}_i$ follows a standard normal distribution. The squared standardized difference $Q_i = \hat{\delta}_i^2$, which is often refered to as the $Q$-statistic, follows a $\chi^2(1)$ under $H_0$. Cochran's $Q$-test for heterogeneity between two studies uses this last property [34,35]. In the replication setting, the $Q$-test measures the evidence that the observed differences between original and replication studies are due to more than just random variation. For an overall test of heterogeneity in a replication project, encompassing $n$ independent study-pairs, the following test statistic and property can be used:

$$Q = \sum_{i=1}^{n} Q_i \overset{H_0}{\sim} \chi^2(n). \tag{2}$$

As discussed above, between-study-pair heterogeneity can be quantified with a *multiplicative* or *additive* variance inflation parameter [12]. For the *multiplicative* heterogeneity parameter $\varphi$, the differences $\Delta_i$ for all study-pairs $i$ are assumed to be independently distributed, such that

$$\Delta_i \sim \mathcal{N}(\Delta, \sigma_{\hat{\Delta}_i}^2 \cdot \varphi). \tag{3}$$

A weighted linear regression model relating the estimated differences $\hat{\Delta}_i$ to a constant is considered and $\varphi$ is estimated as the mean squared error of the model [17,36]. The weights are equal to the inverse of the variance of the difference $\Delta_i$, $w_i = 1/\sigma_{\Delta_i}^2$, ensuring that more precise study-pairs have more influence in the analysis. Further, $\varphi \geq 1$ and in the absence of heterogeneity $\varphi = 1$. In practice, if $\varphi$ is estimated to be smaller than 1, it will be set to 1, as suggested by Mawdsley et al. (2016) [17]. To quantify heterogeneity with an *additive* variance inflation parameter $\tau^2$,

$$\Delta_i \sim \mathcal{N}(\Delta, \sigma_{\hat{\Delta}_i}^2 + \tau^2), \tag{4}$$

a random effects model is considered, with weights assigned to each study-pair $w_i = 1/(\sigma_{\hat{\Delta}_i}^2 + \tau^2)$. The heterogeneity variance $\tau^2$ is then estimated as the between-study-pair variance from this random-effects model, implemented in the `metafor` R-package [16]. In the absence of heterogeneity, $\tau^2 = 0$ and $\tau^2 > 0$ otherwise. The results of both model versions for heterogeneity can be used to compute a prediction interval around the predicted difference in effect size for a new study-pair [20]. The choice between model versions (multiplicative vs additive) depends on the assumptions made about the underlying effect sizes. The multiplicative model assumes a single, common true effect size across study-pairs, whereas the additive model allows for variability in the true effect sizes across study-pairs.

## Location-scale models in replication projects

To investigate sources of shrinkage (i.e., differences) and effect size difference variability between study-pairs (i.e., heterogeneity), meta-regression models for the location and scale will be used. The dependent variable in the regression will be the difference in effect size estimated for each study-pair $\hat{\Delta}_i$ and the units of analysis are the study-pairs $i$. Given a set of $p$ candidate covariates, $x_1, \ldots, x_p$, with values $x_{i1}, \ldots, x_{ip}$ for study-pair $i$, treating heterogeneity as a multiplicative parameter leads to the following model,

$$\Delta_i \sim \mathcal{N}(\beta_0 + \beta_1 x_{i1} + \cdots + \beta_p x_{ip}, \sigma_{\hat{\Delta}_i}^2 \cdot \tilde{\varphi}), \tag{5}$$

where $\tilde{\varphi}$ denotes the residual multiplicative heterogeneity parameter, i.e., the between-study-pair heterogeneity that remains after correcting for the $p$ covariates, $\beta_j$ is the model coefficient for covariate $x_j$, and $\beta_0$ is the model's intercept. For the additive version of heterogeneity, the model is adapted to

$$\Delta_i \sim \mathcal{N}(\beta_0' + \beta_1' x_{i1} + \cdots + \beta_p' x_{ip}, \sigma_{\hat{\Delta}_i}^2 + \tilde{\tau}^2), \tag{6}$$

with $\tilde{\tau}^2$ being the residual additive heterogeneity variance and $\beta_0', \ldots, \beta_p'$ being the intercept and coefficients of the additive model version. Cinar et al. (2021) [21] define the residual heterogeneity as the "variability in the true outcomes not accounted for by the moderators included in the model". The covariates $x_1, \ldots, x_p$ are refered to as location covariates and $\beta_1, \ldots, \beta_p$, respectively $\beta_1', \ldots, \beta_p'$, are the location coefficients, as they stem from an investigation of the covariates' relationship or effect on the location, i.e., the size of the outcome.

As appropriate covariate to explain the amount of shrinkage of effect size, the RPP used the standard error of the original study $\sigma_{oi}$ [1, in Supplement Sect 4.h]. They interpret a positive effect as "imprecise original studies (large standard error) yielding larger differences in effect size between original and replication study" and directly relate a positive effect to evidence for publication bias. Traditionally, the outcome in Egger's regression test [37] is the effect size of each single study. Under the common assumption, that the replication studies are rigorously planned and follow a strict study protocol, and are therefore unbiased, a positive effect of the original standard error on the difference between original and replication study effect sizes can still be related to bias in the original studies.

Location-scale models directly relate covariates to the amount of heterogeneity in the outcome [20]. The scale refers to the heterogeneity, which is now specific to the study-pair. For this, another set of $q$ covariates with values $z_{i1}, \ldots, z_{iq}$ for the $i$th study-pair are introduced, and refered to as "scale covariates". The residual multiplicative heterogeneity parameter $\tilde{\varphi}_i$, respectively the residual additive heterogeneity variance $\tilde{\tau}_i^2$, for original-replication-study-pair $i$ are defined as

$$\ln(\tilde{\varphi}_i) = \alpha_0 + \alpha_1 z_{i1} + \cdots + \alpha_q z_{iq}, \tag{7}$$
$$\text{or } \ln(\tilde{\tau}_i^2) = \alpha_0' + \alpha_1' z_{i1} + \cdots + \alpha_q' z_{iq}, \tag{8}$$

where the $\alpha_0$ and $\alpha_0'$ are the respective intercepts, and $\alpha_k$ and $\alpha_k'$ are the scale coefficients for the scale covariates $z_k$. The log link ensures that the resulting heterogeneity parameters are positive, while $\tilde{\varphi}_i$ additionally is forced to be larger or equal to 1. The models defined in Eqs (5) and (6) are special cases of the location-scale models in Eqs (7) and (8) with $\varphi_i = \exp(\alpha_0)$ or $\tau_i^2 = \exp(\alpha_0')$. The location-scale model with additive heterogeneity is implemented in the R-package metafor [16]. To compute the location-scale model with

multiplicative heterogeneity parameter, generalized additive models for the location, scale and shape, as implemented in the R-package `gamlss` [38], are used with an offset to incorporate weighting. Specifically, combining Eqs (3) and (7), with $\varphi = \tilde{\varphi}_i$, gives $\ln(\sigma_{\hat{\Delta}_i} \cdot \sqrt{\bar{\varphi}_i}) = \ln(\sigma_{\hat{\Delta}_i}) + \frac{1}{2}\ln(\tilde{\varphi}_i)$ leading to an offset of $\ln(\sigma_{\hat{\Delta}_i})$ in the formula for the scale.

**Model selection.** As shown in Cinar et al. (2021) [21], information-theoretic approaches can be used for both, location only and location-scale models. More specifically, to select the most important location covariates, all $2^p$ candidate meta-regression models for the location are considered and their Akaike information criterion (AIC) is computed [39]. AIC is based on the log likelihood $l$ and is defined as AIC $= -2l + 2k$, where $k$ is the total number of model parameters, the number ($p$) of location covariates plus two (intercept and the heterogeneity variance $\tau^2$ or heterogeneity parameter $\varphi$). Note that the likelihood $l$ can be estimated either with maximum likelihood (ML) or restricted maximum likelihood (REML). As shown through simulation studies presented in Cinar et al. (2021) [21] and Viechtbauer and López-López [20], when selecting among candidate models the REML estimation outperformed the ML estimation. Henceforth, we will use REML estimation. The final meta-regression model for the location is the one minimising AIC, including the $\tilde{p}$ best location covariates. To select the $\tilde{q}$ best performing scale covariates among the $q$ candidates, the same selection procedure based on AIC can be used with the exception that $k$ in the definition of the AIC is now equal to the number ($p$) of location covariates plus the number ($q$) of scale covariates plus two (the location intercept and the scale intercept). The same procedure is followed for models with multiplicative and additive heterogeneity.

## Case study: Sources of shrinkage and heterogeneity in psychology and experimental economics

To illustrate the applicability of the proposed methodology, we reanalyse data from the replication projects in Psychology [RPP, 1] and in Experimental Economics [RPEE, 2] provided by Altmejd et al. (2019) [31]. Note that this secondary data analysis is of exploratory rather than confirmatory nature, without any preregistration. The conclusions drawn from our analysis might help formulate new hypotheses to be tested in a subsequent confirmatory study with data collected for purpose.

### Data source and description

Using machine learning models, Altmejd et al. (2019) [31] aimed at predicting reproducibility in two large-scale replication projects (RPP and RPEE) and two many-lab experiments. As outcome measure they used a binary criterion for successful replication defined as a replication with significant effect (two-sided $p$-value $\leq 0.05$) in the same direction as the original study. Additionally, they attempted to predict the relative effect size, i.e., the ratio of replication and original effect sizes, which were standardized to correlation coefficients. The covariates or features used in the machine learning models were divided in two classes: features related to the statistical design properties and outcomes, and features related to the descriptive aspects of the original and replication study, including the citation count of the original articles or the past success of the authors. For our case study, we will employ only RPP and RPEE, since we are interested in between-study-pair heterogeneity which cannot be investigated in many-lab experiments. The data was downloaded from the Open Science Framework (OSF, https://osf.io/4fn73/). As explained in the data analysis protocol in our Appendix, the data provided by Altmejd et al. (2019) [31] was merged with the data on the same replication projects from the R package `ReplicationSuccess` [40]. For our analysis we need standard errors of the effect sizes and are therefore forced to use the so-called meta-analytic

subset for which both the *z*-transformed correlation coefficient and its standard error could be computed (73 of the 100 RPP study-pairs and 18 of the 18 RPEE). Further, Altmejd et al. (2019) [31] included only original studies with an effect interpreted as significant by the original authors (three of the original studies included from the RPP were non-significant). One more study from the RPP was excluded due to too many missing values in the covariates. In total 87 original-replication study-pairs are included in our case study; 69 from the RPP and 18 from the RPEE.

While the machine learning models used all the covariates without any transformation or selection, we base our initial selection on subject knowledge to avoid overfitting given the relatively small sample size and apply log-transformations on count variables. Additionally, we use only information from the original study and information from the replication study that was defined prior to the replication being conducted, which ensures our models remain useful for prediction. In total nine continuous and five categorical covariates were selected as candidates. We refer to our Appendix for a description of the available data and covariates and, specifically, to Table A.3 and Table A.4 for a summary of the selected candidate covariates and applied transformations.

Fig 1 shows the difference in effect size $\hat{\Delta}_i$ for all study-pairs in both replication projects with their 95% confidence intervals. Most replication attempts show high levels of shrinkage of the effect estimate [9]: compared to the original effect size a smaller effect size is observed in the corresponding replication and $\hat{\Delta}_i > 0$. Among all original-replication study-pairs only 14.9% have a negative difference.

## Evidence for shrinkage and between-study-pair heterogeneity

The observed standardized differences $\hat{\delta}_i$ in Eq (1) are computed for all study-pairs. They are shown in the left panel of Fig 2 against a standard normal distribution. In the absence of

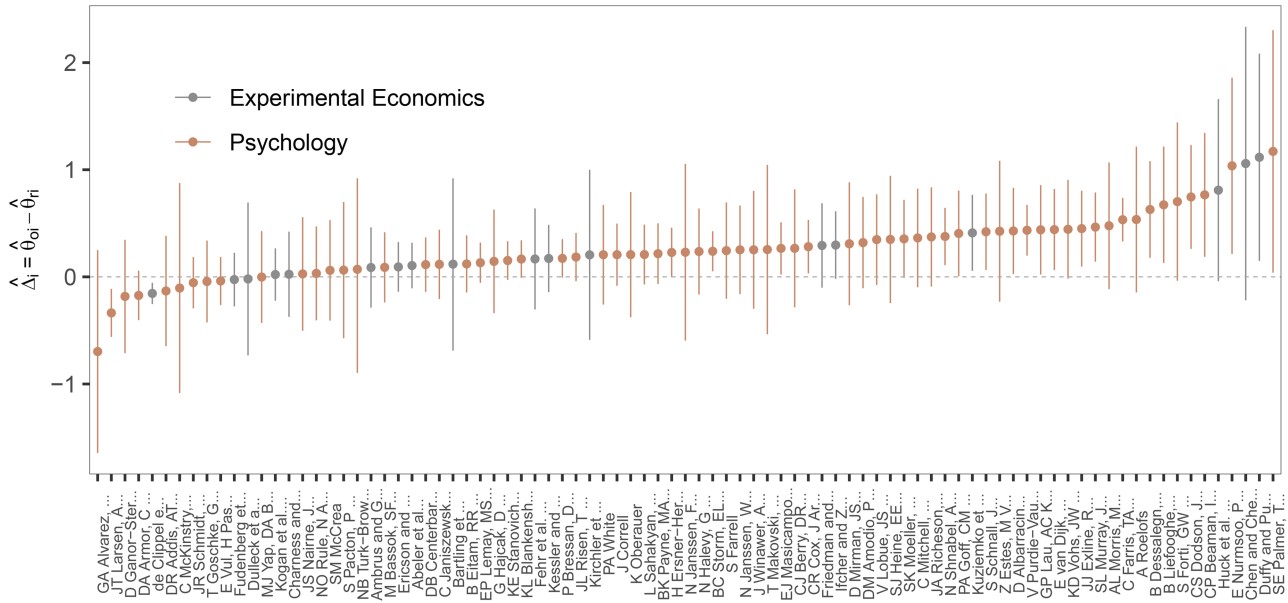

**Fig 1. Ordered differences between effect estimates on Fisher-*z* scale $\hat{\Delta}_i = \hat{\theta}_{oi} - \hat{\theta}_{ri}$ for all included study-pairs ($n = 87$) in both replication projects with their 95% confidence interval.** The dashed horizontal line indicates no difference in effect size.

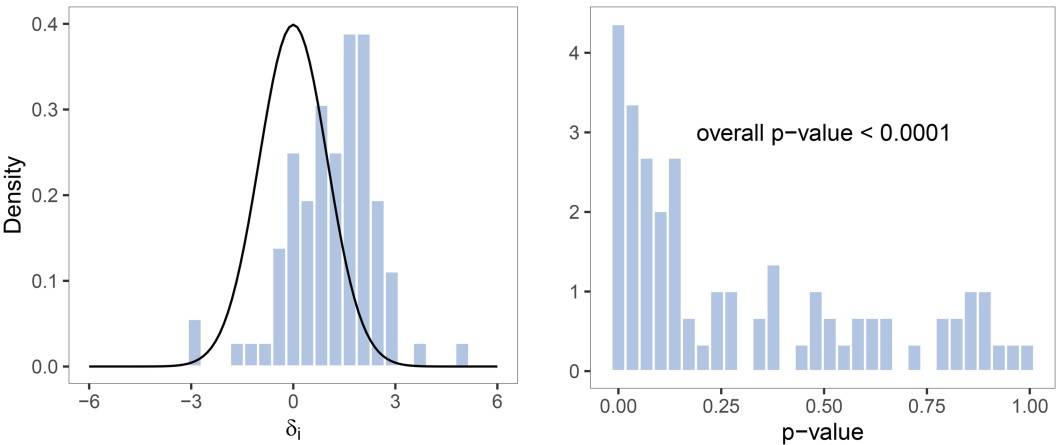

**Fig 2. Left: The distribution of the observed standardized difference $\hat{\delta}_i$ of the original-replication-study-pair** ($n = 87$) **compared to the standard normal distribution. Right**: The $p$-values from the $Q$-test for heterogeneity within original-replication study-pairs, as well as the $p$-value for the overall test for heterogeneity between all study-pairs, included in both replication projects.

shrinkage and heterogeneity, the distribution of the $\hat{\delta}_i$ and the standard normal distribution would be aligned. However, the distribution of the standardized differences is shifted towards positive values, indicating shrinkage. The right panel of Fig 2 shows the $p$-values resulting from a $Q$-test for heterogeneity within each individual original-replication-study-pair. In the absence of shrinkage and heterogeneity, the latter would be uniformly distributed, but too many small $p$-values are observed. An overall test for between-study-pair heterogeneity as described in Eq (2) suggests that heterogeneity cannot be disregarded (overall $p$-value < 0.0001).

The first column of Table 1 shows multiplicative and additive heterogeneity, extracted from the unadjusted models in Eqs (3) and (4) respectively. The estimates of the model intercept can be interpreted as the estimated overall difference in the original-replication comparison. This overall difference is positive, suggesting that, on average the original effect estimates are larger than the effect estimates from the replications, i.e., evidence for shrinkage of effect size. The estimates of the unadjusted heterogeneity presented in Table 1 are the baseline estimates of heterogeneity to be reduced and explained with location-scale meta-regressions. More specifically, using the additive model, the difference in effect size is estimated to be 0.21 with 95% confidence interval from 0.16 to 0.26. The between study-pair variance is estimated to be $\tau^2 = 0.02$ which leads to a 95% prediction interval for the difference in effect size of -0.08 to 0.5 for the difference in effect size for a new study-pair, revealing substantial heterogeneity in the difference between study-pairs.

## Location-only meta-regression models

We follow the analysis presented in the supplement of the RPP [1] and add the standard error of the original study as first location covariate of interest to be included in the meta-regression. We would expect to see more shrinkage for less precise original results, i.e., larger original standard errors. The second column in Table 1 summarises the results of the multiplicative and additive meta-regressions with the resulting residual heterogeneity variance $\tilde{\varphi}$ and residual heterogeneity parameter $\tilde{\tau}^2$ respectively. The results show a positive association between the original standard error and the difference in effect size. As hypothesized, a

**Table 1. Summary of investigated location meta-regression models with multiplicative and additive heterogeneity.** The location coefficient estimates are shown with their 95% confidence intervals (95%CI) and the residual heterogeneity. The first model is the weighted unadjusted meta-regression model relating the difference in effect size to a constant, as in Eqs (3) and (4). For the second model, one covariate was added into the meta-regression as a proof-of-concept. The third model represents the final model selected via the model selection procedure. Additional information on the covariates is added in the note below the table.

| | Unadjusted Model | | Adjusted Model | | Final Model | |
|---|---|---|---|---|---|---|
| | Estimate | 95%CI | Estimate | 95%CI | Estimate | 95%CI |
| **Multiplicative** | | | | | | |
| Intercept | 0.16 | 0.11 to 0.20 | -0.02 | -0.11 to 0.08 | -0.29 | -0.56 to -0.02 |
| Original standard error | — | — | 1.60 | 0.85 to 2.35 | 1.09 | 0.35 to 1.82 |
| Nb authors (O) | — | — | — | — | 0.03 | 0.00 to 0.07 |
| Discipline Econ (ref. Cognitive) | — | — | — | — | -0.28 | -0.46 to -0.10 |
| Discipline Social (ref. Cognitive) | — | — | — | — | -0.02 | -0.13 to 0.09 |
| Nb pages (O) | — | — | — | — | 0.01 | 0.00 to 0.01 |
| O&R same language | — | — | — | — | 0.08 | -0.01 to 0.18 |
| Share male authors (O) | — | — | — | — | -0.17 | -0.31 to -0.04 |
| Avg author citations (log, R) | — | — | — | — | 0.03 | 0.00 to 0.06 |
| Citations (log, O) | — | — | — | — | 0.03 | -0.01 to 0.07 |
| Heterogeneity | $\varphi = 2.004$ | — | $\tilde{\varphi} = 1.682$ | — | $\tilde{\varphi} = 1.212$ | — |
| **Additive** | | | | | | |
| Intercept | 0.21 | 0.16 to 0.26 | 0.06 | -0.05 to 0.17 | 0.09 | -0.11 to 0.28 |
| Original standard error | — | — | 1.11 | 0.36 to 1.86 | 1.07 | 0.35 to 1.79 |
| Nb authors (O) | — | — | — | — | 0.04 | 0.00 to 0.08 |
| Share male authors (O) | — | — | — | — | -0.19 | -0.34 to -0.04 |
| Heterogeneity | $\tau^2 = 0.021$ | — | $\tilde{\tau}^2 = 0.017$ | — | $\tilde{\tau}^2 = 0.012$ | — |

[1]Nb authors (O): number of authors on the original paper.
[2]Discipline: Economics, Social or Coginitive Sciences, with reference being Cognitive Sciences.
[3]Nb pages (O): number of pages of the original paper from the citation infromation.
[4]O&R same language: Experiment of the original and replication study conducted in the same language.
[5]Share male authors (O): proportion of male authors on the original author list.
[6]Avg author citations (log, R): log-transformed average number of citations per author on the replication study.
[7]Citations (log, O): log-transformed number of citations of the original study.

larger $\sigma_{oi}$ (a less precise original finding) leads to larger differences, and hence more shrinkage in effect size ($\beta_1$ is estimated to be 1.6 for the multiplicative model and $\beta_1'$ to be 1.11 for the additive model). The heterogeneity parameter is reduced from $\varphi = 2.004$ to $\tilde{\varphi} = 1.682$ for the multiplicative model and from $\tau^2 = 0.021$ to $\tilde{\tau}^2 = 0.017$ for the additive model. The association between the original standard error an the effect size difference is also shown in Fig B.3 in our Appendix with confidence and prediction intervals depending on whether an additive or a multiplicative version of heterogeneity is used.

More covariates are available to further reduce the residual heterogeneity (see Table A.3 and Table A.4 in our Appendix). A total of 4'608 models depending on which of the covariates are included for the location, additionally to the original standard error. The eight continuous covariates are the original paper length defined as the number of pages from the citation information (according to personal communication with A. Altmejd), the log-transformed citation count of the original study, the number of authors in the original and in the replication team, the share of male authors in the original and in the replication team, and the log-transformed average number of citations per author in the original and in the replication team. The five categorical covariates are the discipline, the highest seniority of the authors in the original and replication team, and binary covariates of the original and replication experiment being conducted in the same language or country, and whether they used the same type of subjects. For the sake of interpretation, the continuous covariates informing on the share

of male authors in the original or the replication study will only be included in combination with the covariates informing on the total number of authors in the original or the replication study, respectively. For both model types, the AIC of the models with best performance (min AIC) per number of covariates is represented in Fig 3, together with the respective residual heterogeneity. The minimum AIC is found after seven more covariates are added in the multiplicative model and two more covariates are added to the additive model. The last column in Table 1 informs on which of the covariates are included in the respective models; see also the description of the included covariates in the notes. Additionally, the residual heterogeneity is shown. The set of included covariates leads to a substantial decrease in heterogeneity: from $\varphi = 2.004$ to $\tilde{\varphi} = 1.212$ for the multiplicative heterogeneity parameter and from $\tau^2 = 0.021$ to $\tilde{\tau}^2 = 0.012$ for the additive heterogeneity variance. Table 1 shows the coefficient estimates and their 95% confidence intervals for the final meta-regression models with only location coefficients. Of interest is that, regardless of the version of the model the magnitude of the coefficient estimates are similar for those covariates included in both models. More authors on the original paper increases the risk of shrinkage (the coefficient in the multiplicative model is 0.03 vs 0.04 in the additive model), while a larger share of male authors in the original author list decreases the effect size difference (the coefficient in the multiplicative is −0.19 vs −0.17 in the additive model). The same model selection steps using Bayesian information criterion can be found in our Appendix. The models with smallest BIC are more parsimounous than those with smallest AIC. The number of original authors and the share of male authors on the original papers are judged most important in both model versions, while the selected multiplicative model with BIC also selects the binary covariate informing on the orignal and replication experiments being conducted in the same country. When selecting with AIC, the covariate "O&R same language", probably highly correlated to "O&R same country" and conveying the same information, was judged more important.

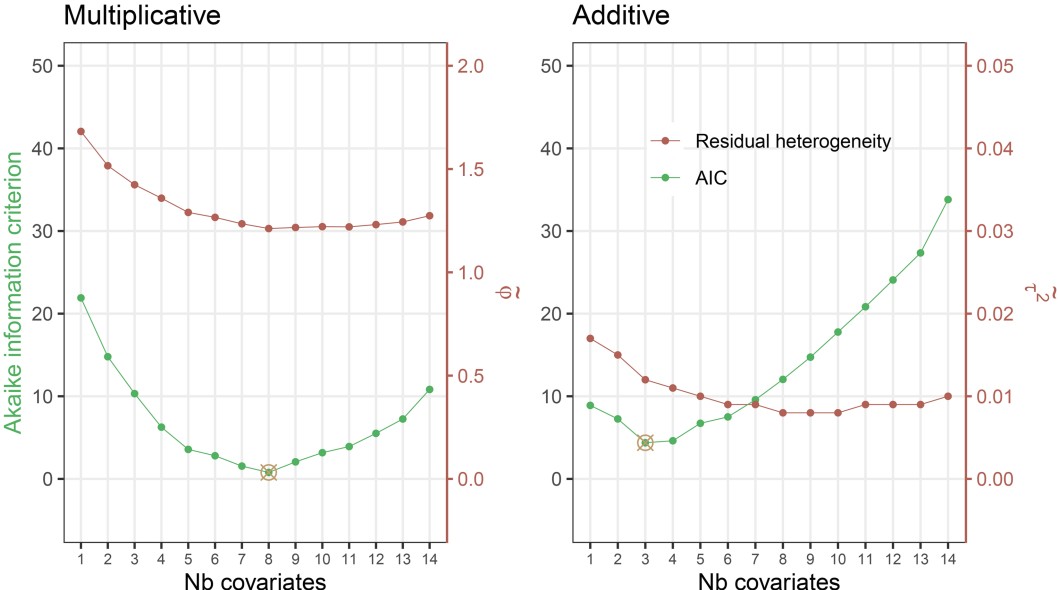

**Fig 3. The AIC for the multiplicative and the additive models with best performance (min AIC) for each possible number of covariates included.** The residual multiplicative heterogeneity parameter $\tilde{\varphi}$ and additive heterogeneity variance $\tilde{\tau}^2$ of the respective models are also shown. At least one covariate, namely the original standard error, is included. The minimum AIC value is highlighted.

## Location-scale meta-regression models

For computational simplicity, only those covariates retained to explain the location in the multiplicative, respectively the additive model, are now also employed as candidates to explain the scale. Tables 2 and 3 show which set of location and scale covariates are included in the best ten models according to their AIC, for the multiplicative and additive heterogeneity respectively. The smallest AIC is observed for the multiplicative model versions by dropping one location covariates (number of pages) and adding the scale covariates "O&R same language" and "Citations (log, O)". For the additive version all location covariates are kept and "Original standard error" is added as scale covariate. The final models are summarised in Table 4. The location coefficient estimates are of similar magnitude as the once presented in Table 1. Turning to the scale covariates, the multiplicative heterogeneity is reduced when the

**Table 2. The ten best multiplicative location-scale models according to their AIC and depending on the selection of location and scale covariates. The location and scale covariates are marked with a check-mark if they are present in the model and with a dash if they are not. The last row of the Table shows the AIC of each model.**

| | Model | | | | | | | | | |
|---|---|---|---|---|---|---|---|---|---|---|
| | 1 | 2 | 3 | 4 | 5 | 6 | 7 | 8 | 9 | 10 |
| **Location covariates** | | | | | | | | | | |
| Original standard error | ✓ | ✓ | ✓ | ✓ | ✓ | ✓ | ✓ | ✓ | ✓ | ✓ |
| Nb authors (O) | ✓ | ✓ | ✓ | ✓ | ✓ | ✓ | ✓ | ✓ | ✓ | ✓ |
| Discipline | ✓ | ✓ | ✓ | ✓ | ✓ | ✓ | ✓ | ✓ | ✓ | ✓ |
| Nb pages (O) | - | - | - | - | - | ✓ | - | - | - | - |
| O&R same language | ✓ | ✓ | ✓ | ✓ | ✓ | ✓ | ✓ | ✓ | ✓ | ✓ |
| Share male authors (O) | ✓ | ✓ | ✓ | ✓ | ✓ | ✓ | ✓ | ✓ | ✓ | ✓ |
| Avg author citations (log, R) | ✓ | - | ✓ | ✓ | - | ✓ | ✓ | - | - | ✓ |
| Citations (log, O) | ✓ | - | - | ✓ | ✓ | ✓ | ✓ | - | - | ✓ |
| **Scale covariates** | | | | | | | | | | |
| Original standard error | - | - | - | - | - | - | - | - | - | - |
| Nb authors (O) | - | - | - | ✓ | - | - | ✓ | ✓ | - | - |
| Discipline | - | - | - | - | - | - | - | - | - | - |
| Nb pages (O) | - | - | - | - | - | - | - | - | ✓ | ✓ |
| O&R same language | ✓ | ✓ | ✓ | ✓ | ✓ | ✓ | ✓ | ✓ | ✓ | ✓ |
| Share male authors (O) | - | - | - | ✓ | - | - | - | ✓ | - | - |
| Avg author citations (log, R) | - | - | - | - | - | - | - | - | - | - |
| Citations (log, O) | ✓ | ✓ | ✓ | ✓ | ✓ | ✓ | ✓ | ✓ | ✓ | ✓ |
| AIC | -11.09 | -10.90 | -10.67 | -9.98 | -9.93 | -9.92 | -9.79 | -9.68 | -9.62 | -9.43 |

**Table 3. The ten best additive location-scale models according to their AIC and depending on the selection of location and scale covariates. The location and scale covariates are marked with a check-mark if they are present in the model and with a dash if they are not. The last row of the Table shows the AIC of the specific model.**

| | Model | | | | | | | | | |
|---|---|---|---|---|---|---|---|---|---|---|
| | 1 | 2 | 3 | 4 | 5 | 6 | 7 | 8 | 9 | 10 |
| **Location covariates** | | | | | | | | | | |
| Original standard error | ✓ | ✓ | ✓ | ✓ | ✓ | ✓ | ✓ | ✓ | ✓ | ✓ |
| Nb authors (O) | ✓ | ✓ | ✓ | ✓ | ✓ | ✓ | ✓ | - | ✓ | ✓ |
| Share male authors (O) | ✓ | ✓ | ✓ | - | ✓ | ✓ | - | - | - | ✓ |
| **Scale covariates** | | | | | | | | | | |
| Original standard error | ✓ | ✓ | - | ✓ | ✓ | - | ✓ | ✓ | - | - |
| Nb authors (O) | - | ✓ | - | - | ✓ | ✓ | ✓ | - | - | ✓ |
| Share male authors (O) | - | - | - | - | ✓ | - | - | - | - | ✓ |
| AIC | 2.25 | 4.06 | 4.37 | 5.01 | 5.17 | 6.24 | 7.01 | 7.26 | 7.26 | 8.09 |

**Table 4. The final multiplicative and additive location-scale meta-regression models minimising the AIC. The location and scale coefficient estimates are shown together with their 95% confidence intervals (95%CI). Only a subset of the covariates used for the location are kept to explain the scale. Additional information on the covariates is added in the note below the table.**

| | Location | | Scale | |
|---|---|---|---|---|
| | Estimate | 95%CI | Estimate | 95%CI |
| **Multiplicative** | | | | |
| Intercept | −0.24 | −0.50 to 0.02 | 1.51 | 0.82 to 2.20 |
| Original standard error | 0.78 | 0.19 to 1.37 | — | — |
| Nb authors (O) | 0.03 | 0.01 to 0.06 | — | — |
| Discipline Econ (ref. Cognitive) | −0.12 | −0.24 to 0.00 | — | — |
| Discipline Social (ref. Cognitive) | 0.07 | −0.02 to 0.16 | — | — |
| O&R same language | 0.12 | 0.02 to 0.21 | −0.44 | −0.76 to −0.12 |
| Share male authors (O) | −0.14 | −0.25 to −0.03 | — | — |
| Avg author citations (log, R) | 0.02 | 0.00 to 0.04 | — | — |
| Citations (log, O) | 0.03 | −0.01 to 0.06 | −0.31 | −0.47 to −0.14 |
| **Additive** | | | | |
| Intercept | 0.09 | −0.1 to 0.29 | -2.02 | -4.89 to 0.86 |
| Original standard error | 1.03 | 0.29 to 1.77 | -26.33 | -59.31 to 6.66 |
| Nb authors (O) | 0.04 | 0 to 0.08 | — | — |
| Share male authors (O) | −0.18 | −0.33 to −0.04 | — | — |

[1] Nb authors (O): number of authors on the original paper
[2] Discipline: Economics, Social or Coginitive Sciences, with reference being Cognitive Sciences
[3] Nb pages (O): number of pages of the original paper from the citation infromation
[4] O&R same language: Experiment of the original and replication study conducted in the same language
[5] Share male authors (O): proportion of male authors on the original author list
[6] Avg author citations (log, R): log-transformed average number of citations per author on the replication study
[7] Citations (log, O): log-transformed number of citations of the original study

experiments in the original and the replication studies are conducted in the same language, and the original study was cited more. The additive heterogeneity is reduced for less precise original studies. The residual heterogeneity resulting from the two location-scale meta-regression models is not constant, but instead a function of the scale coefficients. For study-pairs with average original citation count (on the log scale) of 4.06, the residual multiplicative heterogeneity $\tilde{\varphi} = 1$ for study-pairs using the same language in original and replication experiment, while $\tilde{\varphi} = 1.29$ for those study-pairs with experiments conducted in different languages. Similarly, the residual additive heterogeneity $\tilde{\tau}^2$ is estimated to lie between approximately 0, for maximum original standard errors of 0.58, and close to 0.05 for a minimum original standard errors of 0.04. Graphical model diagnostics of the selected, final location-scale meta-regressions, i.e., normal QQ plots, can be found in Fig C.4 in our Appendix, suggesting that the normality assumption of the data holds.

## Discussion

In this paper, we investigated shrinkage of effect size and heterogeneity in the difference in effect sizes between-original-replication study-pairs that are part of the same large-scale replication effort. We focused on quantifying heterogeneity and explored the potential sources of shrinkage and heterogeneity through covariates associated with study design and demographic factors of the studies and study teams. We suggest to model the differences in effect size between original and replication study with location-scale meta-regression since they allow for a simultaneous investigation of shrinkage and heterogeneity. Traditionally, the unit of analysis in location-scale meta-regressions are individual studies. We extended this

methodology to large-scale RPs by modeling the differences in effect sizes between original and replication study and hence, changing the units of analysis to study-pairs. Location covariates link to the amount of shrinkage of effect size, while the scale covariates help explain heterogeneity in the effect size differences. Commonly used model selection criteria can directly be used on the models to reduce their complexity and select the best covariates for both, the location and the scale [21]. Most literature and methods in meta-analysis and -regression to date focus on quantifying heterogeneity as an additive parameter [32], while we also illustrated how a related model utilizing a multiplicative version of heterogeneity can be computed [17]. Indeed, we believe that the multiplicative heterogeneity has a lot of potential in the setting of RPs due to its easy interpretation and its assumption of a single overall effect size, particularly well suited for direct replication studies based on study protocols that are very similar to those of the original studies.

The results from the location-scale meta-regression models presented in our case study could be of interest to authors of future replication efforts. For example, study-pairs where the experiments of the original and replication study were conducted in the same language present lower multiplicative heterogeneity but more shrinkage. A larger number of authors on the original paper or a longer original paper are associated with more shrinkage for both model versions. However, more precision in the original study (smaller original standard errors) and a larger share of male authors in the original author list are associated with less shrinkage. A larger original standard error, in turn, decreases the additive between-study-pair heterogeneity. Such findings could inform the planning of future replication projects as authors would know how much shrinkage or heterogeneity to expect in effect size differences. In a field where for some reason a lot of shrinkage is to be expected, or where effect sizes generally tend to differ in magnitude from one experiment to the next, traditionally used replication success metrics, which are often based on effect size comparisons, will almost certainly conclude low reproducibility. In these cases, findings from location-scale meta-regression models could, at least, help contextualise the low reproducibility rates, specifically when discussing policies and interventions to improve the reproducibility of research. The findings could further influence the selection of replication success metrics as, some metrics can for example penalise different levels of shrinkage [9].

However, the case study we present is purely observational and any associations that were found could further be confounded by other covariates. In order to confidently base replication project design decisions on such results, large-scale replication projects have to start routinely collecting this information and potentially implement location-scale meta-regression analyses. Additionally, covariates related to questionable research practices or biases in the original study have the potential to be highly informative as sources for shrinkage and heterogeneity. A follow-up study could attempt to specifically collect such information and test the hypothesis that questionable research practices and other biases induce shrinkage and heterogeneity. This is however beyond the scope of our project and would involve experts assessing the risk of bias of the original studies [41]. The expert's risk of bias assessments can then be used to determine whether any remaining effect size differences and heterogeneity exist after bias adjustment [42]. Combining our methodology with a "limit meta-analysis" [43] may constitude an alternative approach to adjust for bias. Note that modeling the heterogeneity using location-scale model is closely related to the framework presented in Holzmeister et al. (2024) [44], where different types of heterogeneity due to differences in study design, population or analysis are isolated and quantified.

Another field of application for the presented methodology is the analysis of heterogeneity across study-pairs of randomised controlled trials (RCTs) and their non-randomised emulations [23]. A meta-regression with only location covariates related to emulation differences

was capable of reducing the heterogeneity substantially. Since shrinkage of effect size is less present in emulations of RCTs, location-scale meta-regression with a special focus on the scale might be worth investigating. There are only limited data on large-scale initiatives comparing RCTs and database studies, none of which have been systematically collecting information to be included in location-scale meta-regressions. The data used in Heyard et al. (2024) [23] is from the RCT-DUPLICATE initiative (Randomized, Controlled Trials Duplicated Using Prospective Longitudinal Insurance Claims: Applying Techniques of Epidemiology) who collected some information on emulation differences but only in a post-hoc manner to be used in a descriptive exploration [45]. More efforts emulating RCTs in real-world data are becoming available [46,47], offering more use cases for our proposed methodology.

## Limitations

Our study is not without limitations. When selecting the models with best prediction performance in our case study, we chose to employ the Akaike information criterion which could have influenced the set of covariates in the final models. Other criteria could be used instead, including the Bayesian information criterion (BIC), scoring rules combined with leave-one-out cross-validation as was done in [23]. We repeated the model selection steps using the BIC (see Appendix) which resulted in more parsimounous models. Since model selection was not the major focus of this project, we refrained from exploring this further. Further, the case study used very limited information on the study-pairs included in two major replication projects. The covariates were collected for their simplicity and convenience, and not for their informative value. As mentioned above, better explantory covariates need to be collected in future RPs. We also suspect some of the covariates to be correlated, as for example the binary indicator informing on the original and replication experiment being conducted in the same language and the binary indicator informing on them being conducted in the same country, which might have influenced our results. Other transformations or combinations of the available covariates could also be use. For example, Pawel and Held (2020) [10] showed how the $z$-statistic of the original study, $z_{oi} = \hat{\theta}_{oi}/\sigma_{oi}$, is associated with shrinkage of effect size. The small number of included study-pairs is another limitation of our case study. Hence, the results should not be over-interpreted but rather spark follow-up studies that use the proposed methodology with data collected for purpose.

## Supporting information

**S1 File. This is our appendix.** It includes data descriptives and supplementary tables, figures and analyses.
(PDF)

## Author contributions

**Conceptualization:** Rachel Heyard, Leonhard Held.

**Data curation:** Rachel Heyard.

**Formal analysis:** Rachel Heyard.

**Methodology:** Rachel Heyard, Leonhard Held.

**Software:** Rachel Heyard.

**Supervision:** Leonhard Held.

**Visualization:** Rachel Heyard.

**Writing – original draft:** Rachel Heyard.

**Writing – review & editing:** Rachel Heyard, Leonhard Held.

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
