## [Decision Letter · Decision Letter 0]

16 Apr 2025

PONE-D-25-08411Meta-regression to explain shrinkage and heterogeneity in large-scale replication projectsPLOS ONE

Dear Dr. Heyard,

Thank you for submitting your manuscript to PLOS ONE. After careful consideration, we feel that it has merit but does not fully meet PLOS ONE’s publication criteria as it currently stands. Therefore, we invite you to submit a revised version of the manuscript that addresses the points raised during the review process.

We look forward to receiving your revised manuscript.

Kind regards,

Yazhou He

Academic Editor

PLOS ONE

Journal Requirements:

Reviewers' comments:

Reviewer's Responses to Questions

**Comments to the Author**

1. Is the manuscript technically sound, and do the data support the conclusions?

Reviewer #1: Yes

Reviewer #2: Yes

Reviewer #3: Yes

Reviewer #4: Yes

Reviewer #5: Yes

Reviewer #6: Yes

2. Has the statistical analysis been performed appropriately and rigorously? 

Reviewer #1: Yes

Reviewer #2: Yes

Reviewer #3: Yes

Reviewer #4: Yes

Reviewer #5: Yes

Reviewer #6: Yes

3. Have the authors made all data underlying the findings in their manuscript fully available?

Reviewer #1: Yes

Reviewer #2: Yes

Reviewer #3: Yes

Reviewer #4: Yes

Reviewer #5: Yes

Reviewer #6: Yes

4. Is the manuscript presented in an intelligible fashion and written in standard English?

Reviewer #1: No

Reviewer #2: No

Reviewer #3: Yes

Reviewer #4: Yes

Reviewer #5: Yes

Reviewer #6: Yes

5. Review Comments to the Author

Reviewer #1: Review of the Manuscript ID PONE-D-25-08411 Title: “Meta-regression to explain shrinkage and heterogeneity in large-scale replication projects” for the Plos One Journal.

General Comments

From my point of view, it is a very interesting topic and simultaneously it seems that to the best of my knowledge is an empirical research aims to use of location-scale meta-regressions as a means to directly relate the identified characteristics with shrinkage (represented by the location) and the heterogeneity variance (represented by the scale). This could also provide valuable insights into drivers and factors associated with high or low reproducibility rates and therefore contextualise results of PRs. The proposed methodology is illustrated using data from the Replication Project Psychology and the Replication Project Experimental Economics. All analysis scripts and data are available online.

The paper consists of following sections: Introduction, Methods, Results, Discussion, Implications for Nursing Education, Limitations, Conclusion.

However, I find some recommendations:

1. The Manuscript needs careful English proofreading because there are some shortcomings. For instance, the article “the” is sometimes missing in front of nouns, the message in some paragraphs is not clear enough. It looks like the first part was written by one author with a greater command of the English language, and the rest of the paper was written by someone else. The numerous grammar errors made this a difficult paper to read. It was strange to see the authors refer to tables that were not submitted. I was unable to find any supplementary material to the submission, so I think this was truly omitted by the authors. Please read the manuscript carefully.

2. It would be very useful to add in the "Introduction" section the purpose, objectives and hypothesis of the research. I consider that a weak point of the paper is that the authors did not show the novelty of the paper compared to other works. That is why, I consider that the introduction should specify the novelty of the paper compared to other papers published in this area.

3. The authors must also show the values of the adjusted R-square, which is more relevant in the methods used in this paper.

4. Authors must present the results of the processing in tabular form and interpret the results. The paper cannot be accepted in this form.

5. The conclusions at the end of the paper should be expanded showing the policy implications of the research results.

In conclusion, the article should be improve. It should also be enhanced with a review of the literature adequate to the subject and a broader interpretation and commentary of the research results.

Reviewer #2: Review of the Manuscript ID PONE-D-25-08411 Title: “Meta-regression to explain shrinkage and heterogeneity in large-scale replication projects” for the Plos One Journal.

General Comments

From my point of view, it is a very interesting topic and simultaneously it seems that to the best of my knowledge is an empirical research aims to use of location-scale meta-regressions as a means to directly relate the identified characteristics with shrinkage (represented by the location) and the heterogeneity variance (represented by the scale). This could also provide valuable insights into drivers and factors associated with high or low reproducibility rates and therefore contextualise results of PRs. The proposed methodology is illustrated using data from the Replication Project Psychology and the Replication Project Experimental Economics. All analysis scripts and data are available online.

The paper consists of following sections: Introduction, Methods, Results, Discussion, Implications for Nursing Education, Limitations, Conclusion.

However, I find some recommendations:

1. The Manuscript needs careful English proofreading because there are some shortcomings. For instance, the article “the” is sometimes missing in front of nouns, the message in some paragraphs is not clear enough. It looks like the first part was written by one author with a greater command of the English language, and the rest of the paper was written by someone else. The numerous grammar errors made this a difficult paper to read. It was strange to see the authors refer to tables that were not submitted. I was unable to find any supplementary material to the submission, so I think this was truly omitted by the authors. Please read the manuscript carefully.

2. It would be very useful to add in the "Introduction" section the purpose, objectives and hypothesis of the research. I consider that a weak point of the paper is that the authors did not show the novelty of the paper compared to other works. That is why, I consider that the introduction should specify the novelty of the paper compared to other papers published in this area.

3. The authors must also show the values of the adjusted R-square, which is more relevant in the methods used in this paper.

4. Authors must present the results of the processing in tabular form and interpret the results. The paper cannot be accepted in this form.

5. The conclusions at the end of the paper should be expanded showing the policy implications of the research results.

In conclusion, the article should be improve. It should also be enhanced with a review of the literature adequate to the subject and a broader interpretation and commentary of the research results.

Reviewer #3: The introduction is a situational context of replication studies (RPs) in psychology and allied sciences. It reports low reproducibility rates and systematic shrinkage of effect sizes. The authors support the use of meta-regression techniques, specifically location-scale models, for exploring drivers of heterogeneity and shrinkage between replication study-pairs.

• Introduction would benefit from a clearer definition of how existing meta-analytic approaches lack an adequate explanation of heterogeneity and how location-scale meta-regression adds to the debate.

• A very concise summary of the data and context (RPP and RPEE) would improve background to aims flow.

• Establish key terms like shrinkage and heterogeneity at the outset for clarity to a multidisciplinary readership.

Literature Review

The article discusses pertinent replication studies (e.g., RPP, RPEE) and presents key statistical techniques from the meta-analysis literature (e.g., additive vs. multiplicative models, location-scale meta-regression).

• Literature review is discontinuous and interrupted by introduction and methodology sections. Perhaps consolidate it all in one section to:

• Strongly distinguishes conceptual contributions of previous meta-regression applications (e.g., RPP supplement, Altmejd et al.).

• Highlights any shortfall in the use of scale modeling or quantification of residual heterogeneity by past studies.

• Merge the rationale for using multiplicative vs. additive heterogeneity models with citations to past applications, explaining when to use each in RP contexts.

Methodology

The authors employ location-scale meta-regression models to replication-minus-original differences in effect sizes across 87 study-pairs in two large-scale replication projects (RPP and RPEE). Both multiplicative and additive heterogeneity models are estimated.

• Make assumptions explicit and test them: While the modeling framework is accurate, the paper can be enhanced by making important assumptions (e.g., normality of residuals, independence of study-pairs) explicit and testing them.

• Residual diagnostics (e.g., Q-Q plots, heteroscedasticity tests) should be described or displayed, particularly for the models in the end.

• Model selection transparency: While model selection using AIC is treated, it would be helpful to justify the specific criteria used (e.g., smallest AIC or parsimony). Cross-validation or leave-one-out procedures can be referenced as checks for robustness.

• Details regarding how covariates were chosen (apart from "subject knowledge") would be valuable—did authors use pre-registered selection procedures, and were there any variables dropped due to multicollinearity?

• Covariates such as "number of authors" and "citation count" can probably be correlated; a VIF analysis or correlation matrix would be added to address multicollinearity.

• The weighted regression approach is appropriate, but independent errors are specified, which would be violated with the pairing of original and replication studies. The article could investigate cluster-robust standard errors or even a multilevel model for improved management of the nested structure.

• Provide further details about how to diagnose the distribution of residuals—especially since inference relies on normality.

• Consider adding goodness-of-fit statistics or R²-type measures, where available, to supplement AIC and facilitate interpretation.

• There could be value in exploring Bayesian meta-regression or reporting bootstrap confidence intervals, especially given the comparatively modest sample size and model complexity.

Results

The results reveal robust evidence of shrinkage (original effects > replication effects) and between-study-pair heterogeneity. Some covariates such as original standard error, study discipline, author gender ratio, and replication language are associated with shrinkage or heterogeneity.

• Coefficient interpretation (e.g., in log-heterogeneity scale) may be difficult to apply for readers. Clarify scale predictors' implications in easy language, maybe using pictorial examples.

• It would help in discussing more narrative around significant results (e.g., the gender makeup finding, the language of replication) and to discuss potential mechanisms or theoretical implications.

• Incorporate a sensitivity analysis, for instance, excluding outliers or re-running analyses with a subset of more similar studies.

• The argument would be assisted by clearer differentiation between correlation and causation. For example, the correlation between proportion of author gender and shrinkage is intriguing, but caution should be taken in attributing such effects to causality.

• Some conclusions (e.g., about precision or experience of authors) would be bolstered by supportive references or theoretical rationale.

• The authors acknowledge the exploratory nature of the study, but this could be even more emphatically stated, particularly with regard to the data-driven model choice.

Reviewer #4: Thank you for the opportunity to review your manuscript titled "Meta-regression to explain shrinkage and heterogeneity in large-scale replication projects." Your study addresses a timely and important issue within the metascientific domain—specifically, how effect size shrinkage and heterogeneity manifest in large-scale replication efforts. The use of meta-regression to explore predictors across multiple replication datasets is both novel and meaningful, and the overall structure of the manuscript is logically organized and well-motivated.

Reviewer #5: The article represents a significant methodological contribution to the field and has great potential for publication. However, there are some improvements that could be made before publishing it.

1) Although the objectives are implicit throughout the text, they could be presented in a more explicit and structured way in the introduction, making it easier to read for a wider, interdisciplinary audience.

2) Although the article compares the two heterogeneity models (additive and multiplicative), the theoretical and practical justification for choosing between them could be better substantiated. I recommend the authors include a section that discusses more clearly in which contexts each model is more appropriate and how the results would change if only one of the models were used.

3) Despite the fact that the coefficients of the models are presented with their confidence intervals, a clearer interpretation of the practical impact of each covariate is lacking. For example: What is the expected magnitude of shrinkage for an increase of 1 unit in the average number of citations?

4) I suggest the authors point out some limitations. The absence of variables on methodological bias or questionable practices. The relatively small sample size (n=87), which may affect the stability of models with many covariates.

5) Suggestions: a future article could compare the different model selection criteria.

Reviewer #6: At the outset, let me to highlight that the research study has already been made available over two online pages:

a. https://ideas.repec.org/p/osf/metaar/e9nw2_v1.html

b. https://osf.io/preprints/metaarxiv/e9nw2_v2

The comments pertaining to this research study are listed below:

1. The problem statement has been well-defined. It clearly identifies the gaps in scientific knowledge and provides strong justification for the current research.

2. The Literature Review has been done effectively. Most of the previous works have been appropriately referenced. It justifies the scope of work undertaken in this research article.

3. A clear understanding of the work undertaken. The researchers have well conceptualized and summarized the entire research work undertaken by them.

4. The researchers have well demonstrated the methodology. The methodology both for shrinkage and heterogeneity used, clearly identifies relevant strengths and weaknesses of the methods.

5. The results are interpreted in light of the proposed research problem and existing literature. This includes explanations and instructional tables and figures as well as supplementary information provided in the appendix. The analysis and Interpretations based on the results are quite convincing.

6. The references are correctly mentioned and are quite adequate in number. The latest references have been provided.

7. There is one suggestion which can somehow ensure the robustness of results pertaining to model selection criterion. In addition to utilizing Akaike information criterion (AIC), other criteria such as Schwarz/Bayesian information criteria or/and scoring rules combined with leave-one-out cross-validation may also be used for presenting respective tables and figures. By doing this, it will also overcome one of the limitation highlighted in the research article.

6. PLOS authors have the option to publish the peer review history of their article (what does this mean?). If published, this will include your full peer review and any attached files.

Reviewer #1: No

Reviewer #2: No

Reviewer #3: **Yes: **Stefanos Balaskas

Reviewer #4: **Yes: **Baranidharan Subburayan

Reviewer #5: **Yes: **Raul Afonso Pommer-Barbosa

Reviewer #6: **Yes: **Muhammad Abdus Salam

---

## [Author Response · Author response to Decision Letter 1]

30 May 2025

Thank you for reviewing our paper. We hereby submit a revised version of our manuscript. Please find below a point by point response to the comments by the reviewers. We provide a version of our manuscript highlighting the changes we made in blue. The comments of reviewer 1 were repeated twice (reviewer 2) - this is why the first header is targeted at reviewer 1 (and 2).

Reviewer 1 and 2

From my point of view, it is a very interesting topic and simultaneously it seems that to the best of my knowledge is an empirical research aims to use of location-scale meta-regressions as a means to directly relate the identified characteristics with shrinkage (represented by the location) and the heterogeneity variance (represented by the scale). This could also provide valuable insights into drivers and factors associated with high or low reproducibility rates and therefore contextualise results of PRs. The proposed methodology is illustrated using data from the Replication Project Psychology and the Replication Project Experimental Economics. All analysis scripts and data are available online. The paper consists of the following sections: Introduction, Methods, Results, Discussion, Implications for Nursing Education, Limitations, Conclusion. However, I find some recommendations:

1. The Manuscript needs careful English proofreading because there are some shortcomings. For instance, the article “the” is sometimes missing in front of nouns, the message in some paragraphs is not clear enough. It looks like the first part was written by one author with a greater command of the English language, and the rest of the paper was written by someone else. The numerous grammar errors made this a difficult paper to read. It was strange to see the authors refer to tables that were not submitted. I was unable to find any supplementary material to the submission, so I think this was truly omitted by the authors. Please read the manuscript carefully.

Reply: We are very sorry to read that the reviewer could not find the appendix, nor the tables / figures. We checked our original submission and the appendix was submitted as a supplement. The figure labels were all in the text, while the Figures themselves were submitted separately, as per the journal’s author guidelines. We realised however that in the original submission the section numbering disappeared in the PLOS LaTeX template which resulted in the “A” referring to Appendix A disappearing in the main text. This was fixed in the revision. We also carefully reread our article with the reviewer’s comment regarding the English proofreading in mind and adapted the language wherever needed.

2. It would be very useful to add in the "Introduction" section the purpose, objectives and hypothesis of the research. I consider that a weak point of the paper is that the authors did not show the novelty of the paper compared to other works. That is why, I consider that the introduction should specify the novelty of the paper compared to other papers published in this area.

Reply: The purpose and novelty of our research paper was mentioned in the last paragraph of our introduction: “In the present paper, we use and extend methodology from the meta-analysis literature to study shrinkage of effect size and heterogeneity in effect size differences between original-replication study-pairs in the research fields Psychology and Experimental Economics. Location-scale meta-regression will help

identify potential sources of shrinkage and heterogeneity, and allow for a more nuanced conclusion on the reproducibility of the research published in those fields.” As this is a methodological contribution with an illustrative and exploratory case study, we do not have a hypothesis to test. To ensure that the contribution of our paper becomes clearer, we re-organised the introduction a bit. The last two sentences of the first paragraph now clearly explain what comes next, the second paragraph title was updated, and we added a third paragraph title.

3. The authors must also show the values of the adjusted R-square, which is more relevant in the methods used in this paper.

Reply: We are investigating changes in heterogeneity, which is the quantity we are reporting. Adjusted R-squared is closely related to the heterogeneity and would, in our opinion, not add much information. However, we now also show the model selection results using the Bayesian information criterion instead of AIC and added an analysis of the residuals to give our results more context (see Appendix).

4. Authors must present the results of the processing in tabular form and interpret the results. The paper cannot be accepted in this form.

Reply: We presented the analysis on the replication projects as an exploratory case study, illustrating the methodology on a published dataset. Our results were presented in tables (tabular form) and we interpreted them, to the best of our knowledge, in the text while reminding the reader that our results should not be overinterpreted. We understand that the reviewer struggled to find Figures and Tables, but they were submitted with the original submission. We improved the readability of the tables by introducing separate columns for the coefficient estimate and the 95% confidence interval (see Tables 1 and 4). We further improved the table descriptions in their captions.

5. The conclusions at the end of the paper should be expanded showing the policy implications of the research results.

Reply: In the second paragraph of our Discussion we explained the relevance and implications of our results for our target audience: authors of future replication efforts. We now mention that such findings could help contextualise low reproducibility rates when discussing policies and interventions to improve the reproducibility of research: “n these cases, findings from location-scale meta-regression models could, at least, help contextualise the low reproducibility rates, specifically when discussing policies and interventions to improve the reproducibility of research”.

Reviewer 3 - Stefanos Balaskas

The introduction is a situational context of replication studies (RPs) in psychology and allied sciences. It reports low reproducibility rates and systematic shrinkage of effect sizes. The authors support the use of meta-regression techniques, specifically location-scale models, for exploring drivers of heterogeneity and shrinkage between replication study-pairs.

- Introduction would benefit from a clearer definition of how existing meta-analytic approaches lack an adequate explanation of heterogeneity and how location-scale meta-regression adds to the debate.

- A very concise summary of the data and context (RPP and RPEE) would improve background to aims flow.

- Establish key terms like shrinkage and heterogeneity at the outset for clarity to a multidisciplinary readership.

Reply: The text of the introduction was slightly adapted to help the flow and understanding. We further changed the title of the second subsection in the introduction, and added a third one, as it might not have been clear before that the last part of the introduction was explaining how location-scale meta-regressions could fill this gap. Shrinkage was in the abstract (“substantial degrees of shrinkage of effect size, where the replication effect size was found to be, on average, much smaller than the original effect size”), and also later in the start of the introduction (“The replication effect sizes were found to be significantly smaller in absolute value than the original effect sizes. This phenomenon is commonly known as shrinkage”). The meaning of heterogeneity was clarified in the abstract: “This often results in between-study-pair heterogeneity, i.e., variation in effect size differences across study-pairs that goes beyond expected statistical variation.” The same phrasing is used in the introduction.

Literature Review: The article discusses pertinent replication studies (e.g., RPP, RPEE) and presents key statistical techniques from the meta-analysis literature (e.g., additive vs. multiplicative models, location-scale meta-regression).

Literature review is discontinuous and interrupted by introduction and methodology sections. Perhaps consolidate it all in one section to:

- Strongly distinguishes conceptual contributions of previous meta-regression applications (e.g., RPP supplement, Altmejd et al.).

- Highlights any shortfall in the use of scale modeling or quantification of residual heterogeneity by past studies.

- Merge the rationale for using multiplicative vs. additive heterogeneity models with citations to past applications, explaining when to use each in RP contexts.

Reply: We believe that the literature mentioned in the introduction summarises past efforts, while the references mentioned in the methods section facilitate our notation and methodology extensions. We hope that the adjustments in the structure of the introduction help the flow of the argumentation. The rationale for using multiplicative other additive heterogeneity models was made more explicit now in the methods section already, right after the models were introduced: “The choice between model versions (multiplicative vs additive) depends on the assumptions made about the underlying effect sizes. The multiplicative model assumes a single, common true effect size across study-pairs, whereas the additive model allows for variability in the true effect sizes across study-pairs.”

Methodology: The authors employ location-scale meta-regression models to replication-original differences in effect sizes across 87 study-pairs in two large-scale replication projects (RPP and RPEE). Both multiplicative and additive heterogeneity models are estimated.

- Make assumptions explicit and test them: While the modeling framework is accurate, the paper can be enhanced by making important assumptions (e.g., normality of residuals, independence of study-pairs) explicit and testing them.

Reply: The assumptions on the RP data are now explicitly mentioned in the first section of the Methods when the notation is introduced, e.g., independence of the included study-pairs.

- Residual diagnostics (e.g., Q-Q plots, heteroscedasticity tests) should be described or displayed, particularly for the models in the end.

Reply: We added QQ plots as model diagnostics of the final location-scale meta-regression models in the appendix (and mention / refer to them in the appendix).

- Model selection transparency: While model selection using AIC is treated, it would be helpful to justify the specific criteria used (e.g., smallest AIC or parsimony). Cross-validation or leave-one-out procedures can be referenced as checks for robustness.

Reply: In our Appendix, we now show the results from the same steps for model selection with BIC (Bayesian information criterion). The resulting models are more parsimonious, including less covariates. However, the covariates are added in the same order of importance. The selection with BIC favours the binary covariate indicating that original and replication experiments are conducted in the same country, while the selection with AIC favours the binary covariate indicating that both experiments are conducted in the same language (these covariates are potentially highly correlated and convey the same information).

- Details regarding how covariates were chosen (apart from "subject knowledge") would be valuable—did authors use pre-registered selection procedures, and were there any variables dropped due to multicollinearity?

Reply: We did not preregister any of our analysis. We now explicitly state this in the manuscript: “Note that this secondary data analysis is of exploratory rather than confirmatory nature, without any preregistration”.

- Covariates such as "number of authors" and "citation count" can probably be correlated; a VIF analysis or correlation matrix would be added to address multicollinearity.

Reply: We agree with the reviewer that some of the covariates might be correlated, and added this in the discussion of the limitations of our study: “We also suspect some of the covariates to be correlated, as for example the binary indicator informing on the original and replication experiment being conducted in the same language and the binary indicator informing on them being conducted in the same country, which might have influenced our results.”

- The weighted regression approach is appropriate, but independent errors are specified, which would be violated with the pairing of original and replication studies. The article could investigate cluster-robust standard errors or even a multilevel model for improved management of the nested structure.

Reply: The original and replication study are supposed to be independent, as they both collect their own data: “Original and replication study are assumed to be two independent studies, each collecting its own data.” We made this even clearer now when introducing the notation.

- Provide further details about how to diagnose the distribution of residuals—especially since inference relies on normality.

Reply: As mentioned above, we added QQ- plots for diagnostics in the Appendix, and mention and interpret them in the main text.

- Consider adding goodness-of-fit statistics or R²-type measures, where available, to supplement AIC and facilitate interpretation.

Reply: As mentioned above, we added the selection based on BIC. Additionally, for each model we discuss and present its (residual) heterogeneity variance, which is related to the R².

- There could be value in exploring Bayesian meta-regression or reporting bootstrap confidence intervals, especially given the comparatively modest sample size and model complexity.

Reply: We agree, but refrain from adding another type of model as we believe the manuscript is already complicated enough as is. It might be of interest for future work, and thank the reviewer for this suggestion.

Results: The results reveal robust evidence of shrinkage (original effects > replication effects) and between-study-pair heterogeneity. Some covariates such as original standard error, study discipline, author gender ratio, and replication language are associated with shrinkage or heterogeneity.

- Coefficient interpretation (e.g., in log-heterogeneity scale) may be difficult to apply for readers. Clarify scale predictors' implications in easy language, maybe using pictorial examples.

- It would help in discussing more narrative around significant results (e.g., the gender makeup finding, the language of replication) and to discuss potential mechanisms or theoretical implications.

- Incorporate a sensitivity analysis, for instance, excluding outliers or re-running analyses with a subset of more similar studies.

- The argument would be assisted by clearer differentiation between correlation and causation. For example, the correlation between proportion of author gender and shrinkage is intriguing, but caution should be taken in attributing such effects to causality.

- Some conclusions (e.g., about precision or experience of authors) would be bolstered by supportive references or theoretical rationale.

- The authors acknowledge the exploratory nature of the study, but this could be even more emphatically stated, particularly with regard to the data-driven model choice.

Reply: As stated before, our case study is purely exploratory, and illustrative of the methodology. We do not make any causal claims and also refrain from hypothesising or making theoretical rationales of certain directions and magnitudes of effects. However, while revising our manuscript, we changed the wording a bit in the discussion of the results.

Reviewer 4 - Baranidharan Subburayan

The manuscript titled "Meta-regression to explain shrinkage and heterogeneity in large-scale replication projects", here is a professional reviewer report tailored to the essential criteria provided for reviewing a manuscript for PLOS ONE.

1. Scientific Validity: The manuscript addresses a significant question in the field of meta-science and psychology the explanation of shrinkage and heterogeneity observed in large

---

## [Decision Letter · Decision Letter 1]

23 Jun 2025

Meta-regression to explain shrinkage and heterogeneity in large-scale replication projects

PONE-D-25-08411R1

Dear Dr. Heyard,

We’re pleased to inform you that your manuscript has been judged scientifically suitable for publication and will be formally accepted for publication once it meets all outstanding technical requirements.

Kind regards,

Yazhou He

Academic Editor

PLOS ONE

Additional Editor Comments (optional):

Reviewers' comments:

Reviewer's Responses to Questions

**Comments to the Author**

1. If the authors have adequately addressed your comments raised in a previous round of review and you feel that this manuscript is now acceptable for publication, you may indicate that here to bypass the “Comments to the Author” section, enter your conflict of interest statement in the “Confidential to Editor” section, and submit your "Accept" recommendation.

Reviewer #1: All comments have been addressed

Reviewer #2: All comments have been addressed

Reviewer #3: All comments have been addressed

Reviewer #5: All comments have been addressed

Reviewer #6: All comments have been addressed

2. Is the manuscript technically sound, and do the data support the conclusions?

Reviewer #1: Yes

Reviewer #2: Yes

Reviewer #3: Yes

Reviewer #5: Yes

Reviewer #6: Yes

3. Has the statistical analysis been performed appropriately and rigorously? 

Reviewer #1: Yes

Reviewer #2: Yes

Reviewer #3: Yes

Reviewer #5: Yes

Reviewer #6: Yes

4. Have the authors made all data underlying the findings in their manuscript fully available?

Reviewer #1: Yes

Reviewer #2: Yes

Reviewer #3: Yes

Reviewer #5: Yes

Reviewer #6: Yes

5. Is the manuscript presented in an intelligible fashion and written in standard English?

Reviewer #1: Yes

Reviewer #2: Yes

Reviewer #3: Yes

Reviewer #5: Yes

Reviewer #6: Yes

6. Review Comments to the Author

Reviewer #1: The authors have modified the paper according to the reviewer's recommendations. I agree to accept the paper for publication.

Reviewer #2: All the recommendations have been applied. The paper substantialy improved. I agree to accept the paper for publication.

Reviewer #3: The authors have adequately addressed all my comments and concerns. I believe the paper is ready for publication.

Reviewer #5: After a careful reading of the manuscript, I consider that the authors have responded satisfactorily to the indicated revisions, so I recommend the publication of this manuscript.

Reviewer #6: The authors have duly incorporated the reviewers’ comments where appropriate, ensuring that key concerns and suggestions have been adequately addressed. The manuscript is now in a well-developed form and is suitable for publication.

7. PLOS authors have the option to publish the peer review history of their article (what does this mean?). If published, this will include your full peer review and any attached files.

Reviewer #1: No

Reviewer #2: No

Reviewer #3: **Yes: **Stefanos Balaskas

Reviewer #5: **Yes: **Raul Afonso Pommer Barbosa

Reviewer #6: **Yes: **Muhammad Abdus Salam

---

## [Editor Report · Acceptance letter]

PONE-D-25-08411R1

PLOS ONE

Dear Dr. Heyard,

I'm pleased to inform you that your manuscript has been deemed suitable for publication in PLOS ONE. Congratulations! Your manuscript is now being handed over to our production team.

Kind regards,

on behalf of

Dr. Yazhou He

Academic Editor

PLOS ONE